# Blood meal-induced inhibition of vector-borne disease by transgenic microbiota

Jackie L. Shane [1], Christina L. Grogan [1], Caroline Cwalina[1] & David J. Lampe [1]

Vector-borne diseases are a substantial portion of the global disease burden; one of the deadliest of these is malaria. Vector control strategies have been hindered by mosquito and pathogen resistances, and population alteration approaches using transgenic mosquitos still have many hurdles to overcome before they can be implemented in the field. Here we report a paratransgenic control strategy in which the microbiota of *Anopheles stephensi* was engineered to produce an antiplasmodial effector causing the mosquito to become refractory to *Plasmodium berghei*. The midgut symbiont *Asaia* was used to conditionally express the antiplasmodial protein scorpine only when a blood meal was present. These blood meal inducible *Asaia* strains significantly inhibit pathogen infection, and display improved fitness compared to strains that constitutively express the antiplasmodial effector. This strategy may allow the antiplasmodial bacterial strains to survive and be transmitted through mosquito populations, creating an easily implemented and enduring vector control strategy.

[1] Department of Biological Sciences, Duquesne University, 600 Forbes Avenue, Pittsburgh, PA 15282, USA. Correspondence and requests for materials should be addressed to D.J.L. (email: lampe@duq.edu)

More than half of the world's population is at risk of vector-borne diseases, which make up one-sixth of all global illnesses[1]. Factors such as insecticide resistance, urbanization, globalization, and climate change have caused these diseases to expand and adapt to new environments. Many of the disease causing pathogens are transmitted through blood ingestion by arthropod vectors, the most prevalent of which are mosquitoes[2]. Current preventative measures for these diseases revolve around vector control strategies, most commonly through the use of insecticides. However, increasing resistance to insecticides creates a need to combat vector-borne diseases with new approaches[3–5].

Progress has been made in novel control strategies that involve mosquito population reduction (i.e., decreasing the number of potential vectors), and population alteration (i.e., changing the ability of vectors to transmit pathogens)[6]. Many studies involve the production and release of transgenically modified mosquitoes to accomplish these goals. These engineered mosquitoes cause sterility or death to achieve population reduction or are refractory to pathogen infection causing population alteration[7–9]. However, like traditional sterile insect technique strategies, maintaining population reduction strategies in the field requires continual reintroductions of the transgenic mosquitoes[3,9] until the local population becomes extinct. The implementation of population alteration strategies in the field is, in principle, very challenging due to the difficulty of spreading and sustaining the transgenes throughout large populations of mosquitoes[5]. For example, Aedes aegypti mosquitoes were engineered to be resistant to the dengue virus; however, after 17 generations, all resistance genes had been mutated or silenced[10]. This could be related to the fitness of the transgenic mosquitoes, which may be decreased due to factors such as extra energy costs to confer resistance, any mutations caused by the introduction of the exogenous gene, and inbreeding complications encountered when trying to introduce the gene into the population[11,12]. To overcome this pitfall, gene drive mechanisms are being investigated to drive these transgenes through populations to achieve high allele frequency[8,13]. However, roadblocks remain, including the development of drive resistance in the vector[14,15], problems associated with releasing large numbers of mosquitoes in endemic areas, and ethical impacts for the surrounding communities[9]. These barriers to implementation of transgenic vectors provide a compelling basis for the development of simpler and easier to manipulate transgenic control strategies.

An alternative population alteration technology is paratransgenesis[16]. This strategy involves the genetic modification of the microbiota to affect their host's phenotype[6,16]. A similar strategy has been effective using the bacterium Wolbachia in mosquito species for population reduction as well as alteration[17]. However, when it was introduced into the microbiome, mosquito population dynamics were compromised[18] and barriers to the transmission of the Wolbachia were erected by native bacterial symbionts[19]. Using the natural bacteria of the mosquito in a paratransgenic approach may overcome some of the burdens encountered during implementation of a transgenically-modified insect species while avoiding the deleterious effects caused by using non-native microbes.

One of the most serious vector-borne diseases is malaria, which causes over 430,000 deaths per year[20]. Malaria is caused by parasites in the genus Plasmodium, which have a complex life cycle involving a mosquito vector and a human host[6]. Importantly, a severe population bottleneck occurs at the oocyst stage of the Plasmodium life cycle within the mosquito midgut, reducing the number of parasites to much <1% of the original number ingested[21,22]. Paratransgenesis to reduce the frequency of malaria transmission seeks to close this bottleneck.

Several bacterial species have been proposed for antiplasmodial paratransgenesis that were recovered from the vectors themselves[23–25]. A particularly attractive species is Asaia bogorensis. Asaia is transmitted from mother to offspring[26], persists into adulthood[27], and densely populates the female midgut, larval gut, and reproductive organs of Anopheles mosquitoes[19,25]. Furthermore, Asaia seems to be conserved in the mosquito microbiome, perhaps due to the benefit it generates for larval development[20]. This bacterium colonizes a substantial range of arthropod disease vectors, including Aedes aegypti and Aedes albopictus, vectors of a variety of human viruses such as dengue, chikungunya, zika, etc[19]., and Scaphoideus titanus, a leafhopper vector that transmits grapevine phytoplasma[28]. Potentially, pathogen inhibition engineered in this bacterium could be used to control a variety of vector-borne diseases. Asaia sp. SF2.1 was genetically modified previously to express antiplasmodial effector molecules that were secreted into the midgut[29]. However, constitutive production of these proteins is deleterious to the fitness of these bacterial strains.

Strains of Asaia suitable for release in the field will have to compete with the microbiota already established in the midgut and thus should be as fit as possible if they are expected to persist and spread throughout the mosquito population, even over short time spans. We hypothesized that producing antiplasmodial effector molecules only when Plasmodium is present inside the midgut should lead to increased fitness when compared to strains that express effectors constitutively. This was accomplished in this study by isolating promoters that are activated by the influx of the nutrients found in the blood meal, hereafter referred to as blood meal induced (BMI) promoters. Here we show that conditionally induced antiplasmodial genes expressed by midgut symbiotic bacteria not only significantly reduce the prevalence of Plasmodium infection compared to strains that use constitutive promoters, but also allow the transgenic bacteria to compete more effectively with wild type Asaia and to improved colonization of the mosquito midgut.

## Results

**Isolation of native Asaia BMI promoters.** To uncover BMI promoters from Asaia sp. SF2.1, the sequenced genome[30] was searched for genes homologous to those known to be induced by blood meal-like conditions in other bacterial species. Many of these genes are important in the pathogenesis of infectious bacteria or are used for iron homeostasis (Table 1). Promoter regions of these genes were cloned into the pGLR1 plasmid and used to induce expression of a dual GFP-lux operon[31].

Six of eight putative BMI promoters exhibited conditional expression when the bacteria were exposed to blood in vitro: AGLR1.Hem, AGLR1.HF, AGLR1.SodB, AGLR1.HlyA, AGLR1.HlyC, and AGLR1.Ferr. Each strain was first qualitatively tested on minimal chocolate agar which contains lysed red blood cells. The strains that produced fluorescence only on blood supplemented plates were then evaluated for induction in liquid media, and GFP production was quantified using a fluorescent spectrophotometer. AGLR1.SodB and AGLR1.Ferr showed no significant difference in fluorescence between growth on minimal media versus growth in liquid media supplemented with blood. AGLR1.Hem, AGLR1.HF, AGLR1.HlyA, and AGLR1.HlyC were significantly induced in the blood supplemented cultures (Fig. 1).

Once the conditionality of the promoters was validated in vitro, experiments were performed inside the mosquito midgut to ensure the conditional activity of the promoters was retained. Each bacterial strain was fed to female Anopheles stephensi mosquitoes in a 0.1 $OD_{600}$ dilution sugar meal. The bacteria were allowed to colonize the mosquito for 24 h and the test groups

**Table 1 Homologous BMI promoters in related bacterial species used to make putative blood-meal induced GFP strains of *Asaia***

| BMI *Asaia* strain | Homologous gene | Original species | Primary gene function | *Asaia* Homolog gene ID | Amino acid similarity (%) | Reference |
|---|---|---|---|---|---|---|
| AGLR1.Hem | HmuT | *Yersinia pestis* | Hemin ABC transporter | P792_11975 | 55 | 63 |
| AGLR1.HF[a] | HmuT | *Yersinia pestis* | Hemin ABC transporter | P792_11975 | 55 | 63 |
| AGLR1.HlyA | HlyA | *Helicobacter pylori* | Hemolysin | P792_09125 | 51 | 64 |
| AGLR1.HlyC | HlyC | *Vibrio cholera* | Hemolysin transferase | P792_09655 | 55 | 65 |
| AGLR1.Ferr | TrxR | *Helicobacter pylori* | Oxidative stress protection | P792_12065 | 43 | 66 |
| AGLR1.SodB | SodB | *Escherichia coli* | Oxidative stress protection | P792_08220 | 62 | 67 |
| AGLR1.Bfr | Bfr | *Escherichia coli* | Iron storage | P792_12870 | 74 | 68 |
| AGLR1.AcnA | AcnA | *Escherichia coli* | TCA cycle enzyme | P792_14035 | 76 | 69 |

[a]AGLR1.HF was constructed using the same but greatly shortened promoter region used in AGLR1.Hem

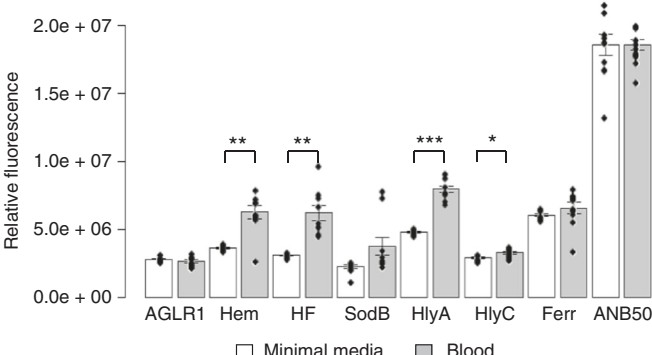

**Fig. 1** Putative BMI promoters significantly increase fluorescence of four *Asaia* conditional strains when exposed to blood. There was a significant difference in mean fluorescence for four BMI isolates in minimal media compared to media supplemented with blood (Welch's $t$-test, $n = 10$ for each strain and condition). No significant difference between the different media was seen for both the positive (ANB50) and negative (AGLR1) controls, as well as the SodB and Ferr strains. Fluorescence readings were taken during log phase and normalized to the $OD_{600}$ of each culture. Height of bars are the mean value for that treatment. Error bars represent standard error of the mean (s.e.m.) ($n = 10$). Individual data points are superimposed on the bar charts. Statistical significance was determined using Welch's two-tailed $t$-test where significance is represented by $*P < 0.05$, $**P < 0.01$, and $***P < 0.001$. Only significant differences are labeled

were split for subsequent blood or sugar feeding. After 2 days all the mosquito midguts were dissected, and each strain was analyzed for differential fluorescence between blood-fed and sugar-fed conditions (Fig. 2). *Asaia* strains AGLR1.Hem, AGLR1. HF, AGLR1.HlyA, and AGLR1.HlyC retained their conditionality in mosquitoes. AGLR1.SodB and AGLR1.Ferr exhibited high background or constitutive expression inside the midgut, respectively, which was similar to their behavior in liquid media.

**Construction and fitness measures of antiplasmodial strains**. The four validated BMI promoters were cloned from the reporter plasmid pGLR1 into the scorpine secretion plasmid pCG18 [https://www.ncbi.nlm.nih.gov/nuccore/?term = MG702576]. A transcriptional terminator was added upstream of the BMI *Asaia* promoters to ensure that there was no transcriptional interference from the kanamycin resistance gene, creating pCG18.glr1 [https://www.ncbi.nlm.nih.gov/nuccore/?term = MG702577] (Fig. 3). The promoters drive a scorpine antiplasmodial effector gene fused to *E. coli* phoA which encodes an alkaline phosphatase reporter gene. This effector configuration was used previously

with constitutive promoters and was shown to be highly refractory to *P. berghei* infection[29]. The BMI scorpine strains were tested for conditionality on blood and were found to produce the scorpine fusion protein only when blood was present (Fig. 4), reproducing the pattern found with GFP.

Before testing for *Plasmodium* inhibition, the fitness of each of the strains was assessed to evaluate whether they differed from the *Asaia* strain that constitutively expresses scorpine. These data were measured in three ways: comparisons of maximum growth rate, competition between wild-type *Asaia* and transgenic strains in co-culture, and the relative ability of the strains to colonize mosquito midguts. The results of these analyses are shown in Fig. 5.

Firstly, the maximum growth rate (μmax) of each strain was measured (Fig. 5a). While none of the paratransgenic strains grew as well as the wild-type *Asaia* control, the ACG18.Hem and ACG18.HlyA strains showed a significant increase in μmax over the constitutive positive control strain ACG18.

Secondly, a competition experiment was performed in which each antiplasmodial strain was grown in the same culture at an equal initial density with wild-type *Asaia*. First, a plasmid retention experiment was conducted for all strains. Over a period of 6 h no significant loss of plasmid occurred while the strains were grown without antibiotic (Supplementary Fig. 1). For the competition, the scorpine secreting strains were grown in a 50/50 co-culture with wild-type *Asaia* inoculated at a 0.5 $OD_{600}$, which is the beginning of log phase for the bacteria. These cultures were allowed to grow for 6 h and then plated on minimal media with or without antibiotic. The ratio of surviving transgenic *Asaia* was calculated by comparing the number of colony forming units (CFUs) on the kanamycin supplemented plates, which only contained strains harboring the antiplasmodial plasmid, to the non-selective plates, containing all of the bacteria present in the culture (Fig. 5b). The constitutively expressing strain showed the greatest reduction in the ratio of paratransgenic to wild-type CFUs while the BMI strains all retained significantly higher ratios of transgenic bacteria compared to the constitutive control strain. This is especially true for the ACG18.Hem construct which represented half of the bacterial population when grown with the wild-type *Asaia* strain, the same proportion as at the beginning of the experiment.

Finally, the relative ability of these strains to colonize mosquito midguts was measured. The transgenic bacteria were fed to the mosquitoes as was done in the fluorescence experiments described above. After 2 days, ten midguts carrying each strain were dissected, homogenized in PBS, and paratransgenic CFUs were enumerated (Fig. 5c). ACG18.HlyA had the highest rate of colonization in the midgut followed by ACG18.Hem. A significant increase in colonization was also seen for ACG18.HF

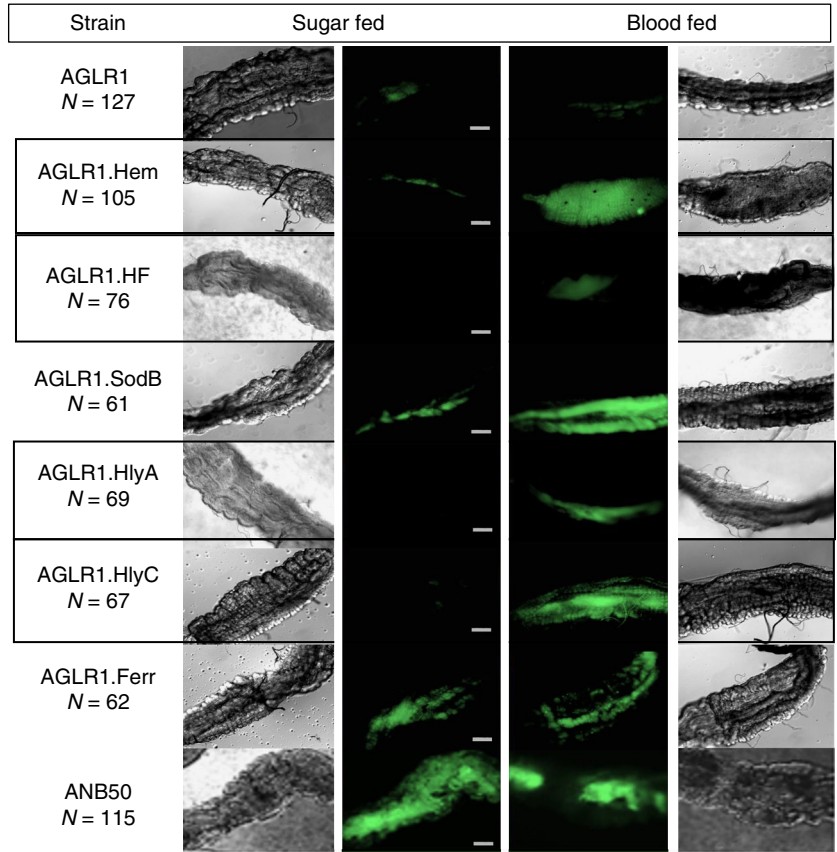

**Fig. 2** BMI *Asaia* strains retain GFP induction in vivo in response to a mosquito blood meal. Bacterial cultures of each strain were fed to mosquitoes and differences in fluorescence were analyzed by eye between the blood fed and sugar fed individuals of each strain. AGLR1, which contains no promoter, is the negative control and ANB50 is a constitutively fluorescent positive control. The strains outlined in gray, AGLR1.Hem, HF, HlyA and HlyC, show conditional fluorescence when exposed to a blood meal vs sugar meal indicating these promoters are induced by the blood meal in the mosquito. Images are representative of all observed midguts and all midguts were observed at the same magnification. The size bars are each 100 μM

compared to ACG18. ACG18.HlyC did not show a significant increase in colonization over the constitutive control.

**BMI promoters used for *Plasmodium* inhibition.** Each anti-plasmodial *Asaia* strain and wild-type *Asaia* were fed to *A. ste-phensi* mosquitoes as was done with the GFP strains. The mosquitoes were then fed on mice infected with *Plasmodium berghei*, a rodent model system for malaria. The parasites were left to develop in the mosquitoes for 12 days before the mosquitoes were dissected and oocyst numbers counted for each midgut (Fig. 6). A significant decrease in median oocyst numbers occurred for all strains secreting the antiplasmodial construct when compared to mosquitoes fed on the wild-type *Asaia* sp. SF2.1. All scorpine secreting strains reduced the oocyst median number to almost one tenth of the wild-type control, indicating a strong antiplasmodial effect in these paratransgenic mosquitoes. Most importantly, the prevalence (the number of midguts showing any oocysts at all) of *P. berghei* infection was reduced significantly (26.3% – 41.4%) for all BMI strains compared to wild-type *Asaia* ($\chi^2$, $P < 0.01$), whereas the constitutive strain did not ($\chi^2$, $P = 0.081$). Three of the four BMI strains reduced prevalence to a significant degree when compared to the constitutive strain. The strain that performed best against *P. berghei* (ACG18. Hem) also scored highest in two of the three fitness measurements (μmax and competition with wild-type *Asaia*), and second highest in its relative ability to colonize the mosquito midgut (Fig. 5).

## Discussion

Mosquito-borne diseases remain a persistent threat to over half of the world's population, and their range has been spreading due to globalization, urbanization, and climate change[32–34]. Current preventative strategies center around controlling the vectors of these diseases; most often, this involves the use of bed nets and insecticides[3,35,36]. However, many mosquito species have evolved resistances to commonly used insecticides[37] and some have even altered their biting behavior to gain access to humans at times of the day when bed nets are not being used[38]. Other vector control strategies involve transgenically modifying the mosquito vector genome; however, mutations in transgenes, reproductively isolated populations, as well as the large number of species that vector particular pathogens may make implementing this technology challenging in the field[6,39,40].

Paratransgenesis offers the possibility to bypass these roadblocks by utilizing bacteria to alter the disease transmission phenotype of mosquito vectors. Several bacterial species have been developed for use in antiplasmodial paratransgenesis, including ones that are naturally antiplasmodial (e.g., *Enter-obacter* sp Esp_Z. and *Chromobacterium* sp. Csp_P.)[23,24] or that have been genetically modified (e.g., *Pantoea agglomerans*, *Ser-ratia marcescens AS1*, and *Asaia* sp. SF2.1)[29,41,42]. An ideal paratransgenic candidate is one that can be genetically modified, colonizes vector mosquitoes in the same body compartments where pathogens develop, spreads through mosquito populations, and is not a human pathogen. *Asaia* sp. SF2.1 meets all of these criteria. It colonizes *Anopheles* midguts, testes, ovaries, and

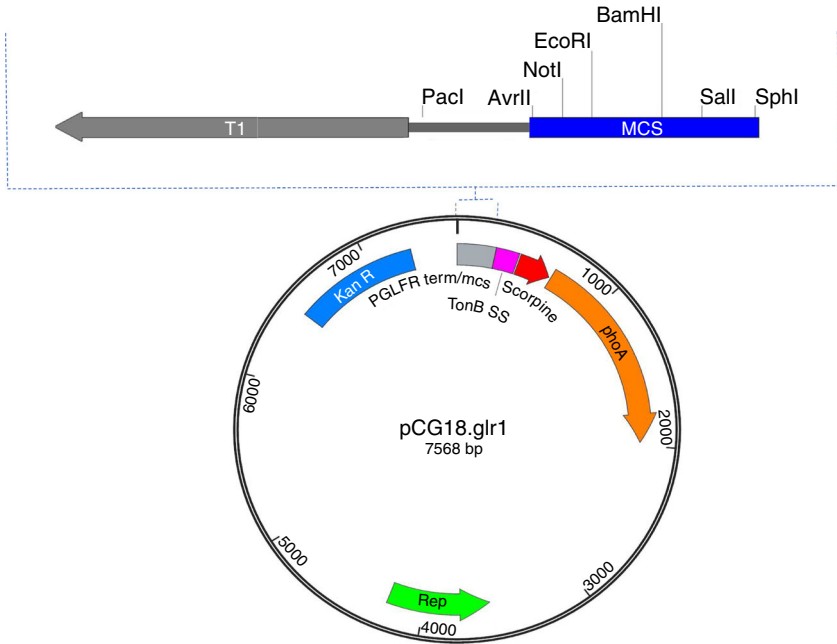

**Fig. 3** Antiplasmodial vector pCG18.glr1. BMI promoters were added at the multiple cloning site to drive the antiplasmodial effector scorpine in the promoterless pCG18.glr1. This plasmid was generated from the fusion of the promoter region of pGLR1 into the constitutive antiplasmodial plasmid pCG18 that contains a secreted protein fusion of scorpine and alkaline phosphatase. T1 transcriptional terminator from *E. coli rrnB*, Rep pBBR1 origin of replication, KanR kanamycin resistance, mcs multiple cloning site, TonB SS *Asaia* TonB dependent signal sequence for secretion, phoA *E. coli* alkaline phosphatase

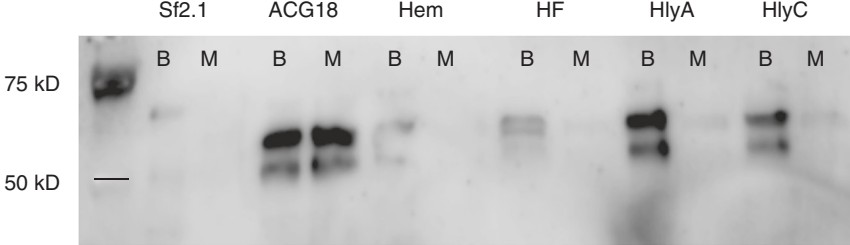

**Fig. 4** Scorpine is conditionally expressed in all BMI isolates when blood is present. Expression of scorpine was evaluated from colonies collected from minimal media agar plates (M) and chocolate agar plates that contain lysed blood (B) in a western analysis using an anti-alkaline phosphatase antibody to detect the fusion constructs. Two bands were detected by the antibody. The larger band (ca. 61.1 kD) correlates with the size of the secreted protein that contains the secretion signal while the smaller band (ca. 55.4 kD) correlates to the size of the protein construct lacking the secretion signal. All conditional constructs show protein production only when blood is supplemented in the media, whereas the constitutive strain (ACG18) produces the fusion protein on both media. SF2.1 = wild type *Asaia*; ACG18 = constitutive expression of scorpine by *Asaia*; all others are strains where scorpine is driven by a BMI promoter

salivary glands[25] and spreads in mosquito populations vertically and horizontally[43]. Importantly, there are very few reported human infections caused by members of the genus *Asaia*[44–48]. *Asaia* is not naturally antiplasmodial, so to make it a useful paratransgenic platform it must be genetically modified to express antiplasmodial factors. Genetic modifications of bacteria, however, can lead to a severe loss of bacterial fitness. In laboratory strains of *E. coli*, for example, numerous strategies have been developed over the years to control heterologous gene expression to allow for the production of foreign proteins because experience has shown that this often leads to a severe loss of fitness[49,50]. Strategies include varying the gene dosage of the heterologous gene by utilizing plasmids that vary in copy number and, more commonly, using conditional promoters to tightly control gene expression to the time preferred by the experimenter. For anti-plasmodial paratransgenic strains to be most effective, they should be as fit as possible in order to compete with the microbiota already present in the midgut, and this will be especially true in the field where we can expect the microbiota of *Anopheles* to be more complex than that of lab colonies. We hypothesized that placing antiplasmodial genes under the control of a conditional promoter would increase *Asaia* fitness, allow the transgenic strains to colonize the midgut more efficiently, and lead to improved paratransgenesis.

The most obvious change of condition in the mosquito related to *Plasmodium* sp. infection is the ingestion of an infected blood meal by female *Anopheles* mosquitoes. The introduction of blood into a mosquito midgut causes a striking environmental shift for the symbiotic bacteria in the midgut as well as a substantial increase in microbial flora[51]. This change in environment can induce a variety of conditionally expressed genes in the native microbiome. We isolated four *Asaia* promoters based on homology to genes in other bacterial species that showed conditional expression when exposed to blood. These promoters

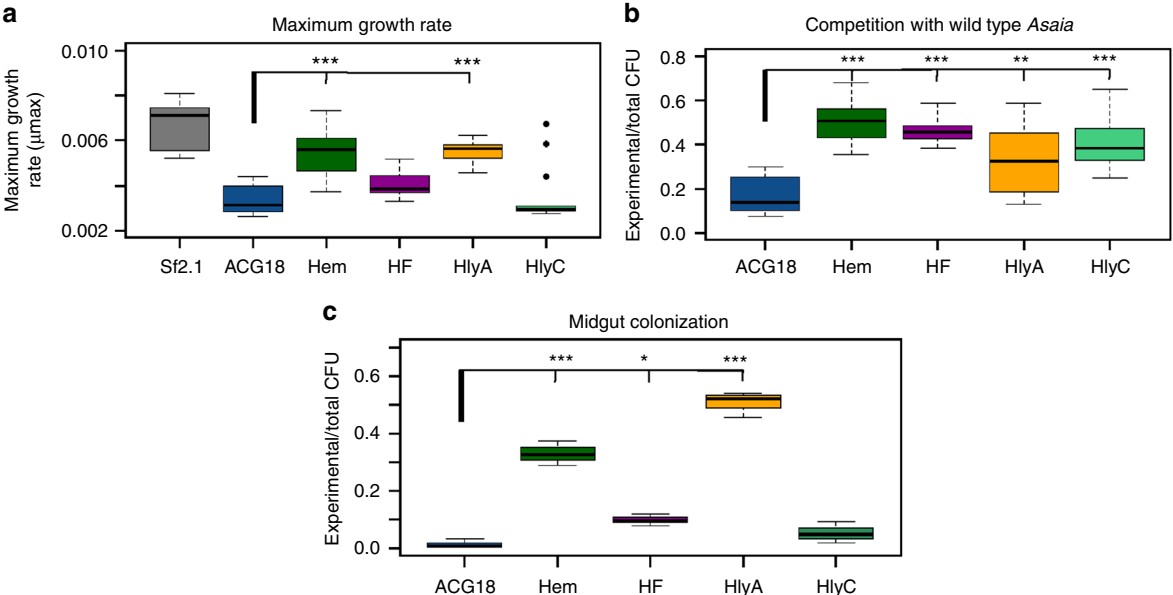

**Fig. 5** BMI strains demonstrate increased fitness compared to a constitutively-expressing strain. **a** Growth curves and maximum growth rates were calculated from 10 individual isolates of each strain grown over log phase of the bacteria using RStudio; **b** Each strain was competed in minimal media with wild-type *Asaia* SF2.1 in an initial 50/50 co-culture during log phase of the bacteria. Ratios of transgenic vs. whole culture CFUs are displayed. A 50:50 ratio indicates no loss of the paratransgenic strain during the course of the experiment; **c** Antiplasmodial *Asaia* strains were fed to mosquitoes and 10 midguts from each sample were pooled and plated on selective media. Transgenic CFUs were counted for each isolate and taken as a ratio of the total across all strains. For each panel, box bars are medians. The top and bottom of the boxes represent the first and third quartile of the data spread. The lower and upper bounds of the whiskers are the lowest datum still within 1.5X interquartile range (IQR) of the lower quartile, and the highest datum still within 1.5X IQR of the upper quartile, respectively. Data lying beyond the upper or lower 1.5X IQR ranges are shown as single points. Statistical significance for each experiment was determined using one way ANOVA with Dunnett's correction where significance is represented by *$P < 0.05$, **$P < 0.01$, and ***$P < 0.001$ with experimental replicates ($n = 10$, **a**; $n \geq 10$, **b**; $n = 3$, **c**). Only significant comparisons are shown

allowed induction of GFP in the midguts of blood fed female *An. stephensi* (Fig. 2) and retained conditional expression of the antiplasmodial peptide scorpine when the bacteria were grown on agar plates that contained lysed red blood cells (Fig. 4). Importantly, the BMI strains showed significant increases in maximum growth rate, ability to compete when co-cultured with wild-type *Asaia*, and in their ability to colonize mosquito midguts when compared to *Asaia* strains where scorpine was constitutively expressed (Fig. 5), all measurements commonly associated with bacterial fitness. Finally, three of the BMI strains led to superior performance when the strains were employed paratransgenically to block *P. berghei* development in *An. stephensi* (Fig. 6). In particular, the BMI paratransgenic strains produced a significant reduction in parasite prevalence when compared to the strain that constitutively expressed scorpine. Since even a single oocyst can make a mosquito infectious, prevalence is the statistic of importance. Indeed, it is likely that *Asaia* would perform even better under more natural conditions, considering that, in nature, mosquitoes are rarely infected with more than 1–5 oocysts per midgut[6,21], while our control mosquitoes had a median number of 19 oocysts per midgut.

Although the BMI strains reported here improve paratransgenesis, these specific strains are not appropriate for use in the field for at least two reasons. Firstly, the effector gene is carried on a broad host range plasmid that needs drug selection in order to be retained by the bacteria. Field strains would necessarily require effector genes to be inserted into the *Asaia* chromosome where they can be inherited stably without drug selection. In addition, chromosomal insertion is a strategy that is far less prone to horizontal gene transfer than are plasmids. Secondly, the strains developed here used a single effector gene to kill parasites in mosquitoes. Field strains would need to express

more than one antiplasmodial effector gene simultaneously to suppress the evolution of resistance by *Plasmodium* to the antiplasmodial products, as situation that is virtually guaranteed if a single effector is used. Nevertheless, the conditional expression of antiplasmodial genes by paratransgenic *Asaia* represents a first step toward improving this species for eventual use in the field, and is a strategy that could be employed in paratransgenic bacterial strains targeted against any blood feeding arthropod disease vector.

## Methods

**Bacterial strains and plasmids**. All plasmid construction was performed in *E. coli* Top10 F′ (Invitrogen). Bacterial cultures were grown at 30 °C for both liquid and solid media. Plasmids were maintained in *E. coli* using 30 mg L$^{-1}$ kanamycin and 120 mg L$^{-1}$ kanamycin in *Asaia*. A list of all plasmids created in this study and the genotypes of the bacterial strains that were used are listed in Supplementary Table 1.

A PCR cloning strategy was used for all GFP-containing pGLR1-derived plasmids. In order to isolate blood-induced promoters in *Asaia*, the genomes of other better-characterized bacterial species were searched for genes shown to be induced when the bacteria encountered blood meal-like conditions. These BMI genes were aligned to the *Asaia sp.SF2.1* genome using BLAST to find homologs (Table 1). The promoter regions of these genes were estimated to be contained within 500 base pairs upstream of the translational start site or up to the next open reading frame if that was encountered first. If the homologous gene was contained in an operon, the promoter region of the entire operon was used. For strains AGLR1.Hem and AGLR1.HF different sizes of the same promoter region was used due to the presence of a gene transcribed on the reverse DNA strand. The promoter region of the operon that contained the HlyA gene was analyzed using Bprom[52] due to its complicated annotation in NCBI. Primers used to amplify promoter regions are listed in Supplementary Table 2.

Each putative BMI *Asaia* promoter was amplified from *Asaia* SF2.1 genomic DNA using a primer pair (Supplementary table 2) specific for the promoter and designed to amplify a region 500 bp upstream of the first ORF in the operon of the homologous genes listed in Table 1.These reactions (and all subsequent PCR reactions used for cloning) used Phusion DNA polymerase (NEB) and the reaction

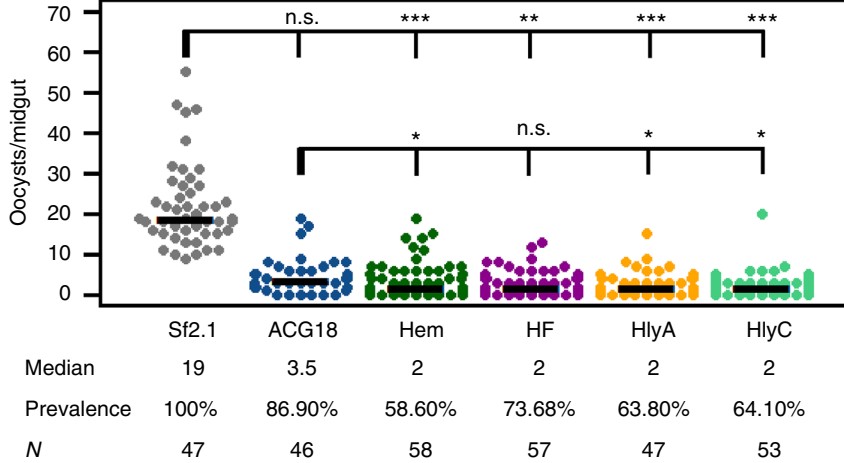

| | Sf2.1 | ACG18 | Hem | HF | HlyA | HlyC |
|---|---|---|---|---|---|---|
| Median | 19 | 3.5 | 2 | 2 | 2 | 2 |
| Prevalence | 100% | 86.90% | 58.60% | 73.68% | 63.80% | 64.10% |
| N | 47 | 46 | 58 | 57 | 47 | 53 |

**Fig. 6** BMI strains significantly reduce the prevalence of *Plasmodium* infection in mosquito midguts compared to the constitutively expressing strain. In three separate trials, oocysts were counted in mosquitoes infected with *Asaia* strains that were fed on a *P. berghei* infected mouse. Each dot represents an individual midgut and the number of *P. berghei* oocysts it contained. Prevalence is the fraction of midguts with at least one oocyst. SF2.1 is the wild type *Asaia* strain (negative control) and ACG18 is the constitutively-expressing positive control. All strains secreting scorpine significantly reduced the median number of oocysts (horizontal bars) compared to the wild type strain (quantile regression, $P \ll 0.001$). There was no significant difference in median oocyst number between the constitutive paratransgenic strain (ACG18) and any of the BMI strains (quantile regression, $P > 0.05$). The prevalence of infection (horizontal comparisons in the figure) was not significantly different between the wild-type and the positive constitutive control, while all of the BMI strains were significantly different ($\chi^2$, 1 df). All of the BMI strains except ACG18.HF showed a significant decrease in prevalence compared to the constitutive positive control ACG18 ($\chi^2$, 1 df). *P*-values: *$P < 0.05$, **$P < 0.01$, ***$P < 0.001$

conditions specified by the manufacturer in a final volume of 25 μl. The reactions followed a 95 °C (1 min), 52 °C (30 s), 72 °C (1 min) cycling profile for a total of 30 cycles. The reactions were cleaned and concentrated using the DNA Clean & Concentrator-5 kit (Zymo Research). Purified PCR products were then cut with *Avr*II and *Sph*I restrictions enzymes and ligated into the same sites of the pGLR1 vector except for the Hem and ferr promoter fragments which were cut by *Eco*R1 and *Bam*H1 and ligated into those same sites in the vector. All ligations followed an optimized temperature-cycle ligation reaction[53].

PCR cloning and Gibson assembly[54] were used to construct the antiplasmodial plasmids. A plasmid that constitutively expressed scorpine under the control of the *nptII* promoter from Tn5 was constructed in two steps. Firstly, pNB92[29] and a GBlock (IDT) synthetic dsDNA fragment (Supplementary Table 2) that contains the first 150 nt of the *Asaia* TonB receptor protein 1 gene were each digested with *Nde*I and *Pac*I, the DNAs purified, and the two ligated together to create pCG6 [https://www.ncbi.nlm.nih.gov/nuccore/?term = MH734181]. Next, pCG6 [https://www.ncbi.nlm.nih.gov/nuccore/?term = MH734181] and pNB97[29] were each cut with *Pac*I and *Sbf*I. The vector backbone-TonB-*phoA* fragment from pCG6 [https://www.ncbi.nlm.nih.gov/nuccore/?term = MH734181] and the scorpine gene fragment from pNB97 were purified and then ligated together to create pCG18 [https://www.ncbi.nlm.nih.gov/nuccore/?term = MG702576].

Plasmids expressing scorpine under the control of BMI promoters were constructed in two steps. Firstly, the primers FpGLR1term/mcsNsiI and RpGLR1term/mcsNdeI (Supplementary Table 2) were used to amplify an upstream transcriptional terminator and multiple cloning site from pGLR1 in a PCR reaction identical to that described above except pGLR1 was used as the template DNA. The PCR product was cleaned and concentrated as described above and the PCR product cut with *Nsi*I and *Nde*I restriction enzymes and ligated into the same sites in pCG18 [https://www.ncbi.nlm.nih.gov/nuccore/?term = MG702576]. This replaced the constitutive promoter with the terminator and multiple cloning sites leading to the creation of pCG18.glr1 [https://www.ncbi.nlm.nih.gov/nuccore/?term = MG702577] (Fig. 3). Next, a two-fragment Gibson assembly procedure was used to introduce specific BMI promoters into pCG18.glr1 [https://www.ncbi.nlm.nih.gov/nuccore/?term = MG702577]. The plasmid backbone was amplified using pCG18.glr1 [https://www.ncbi.nlm.nih.gov/nuccore/?term = MG702577] as a template and the primers F18promGibs and R18promGibs. BMI promoter regions were amplified from their cognate pGLR1-based clones using the BMI promoter specific primer pair listed in Supplementary table 2. Both PCRs were run and the products purified as described above. A two-part Gibson assembly reaction was conducted exactly as described by Gibson et al[54]. the products transformed into *E. coli* Top10F' cells (Invitrogen), and selected using kanamycin.

In all cases, after sequence confirmation, plasmids were transformed by electroporation[55] into *Asaia* sp.SF2.1[25]. Plasmid sequences were deposited in Genbank under accession codes MG702576 though MG702589, and MH734181 for pCG6.

**Mosquito and parasite growth and maintenance**. *Anopheles stephensi* mosquitoes were a gift from the Johns Hopkins Malaria Research Institute. The identity of the mosquitoes was confirmed by using morphological stock identification characters recommended by the Malaria Research and Reference Reagent Resource Center[56]. Mosquitoes were reared at 29°C with a 12 h light:12 h dark photoperiod and maintained as adults on a 10% sucrose solution supplied on cotton wicks. Mated females were fed on anesthetized mice for a blood meal and allowed to lay eggs on wet filter paper for 2 days. All experiments involving mice were approved by the Duquesne University Institutional Animal Care and Use Committee and followed approved ethical practices. Eggs were washed from the filter paper into larval water in photographic trays until the eggs hatched. Larvae were reared at 29 °C and fed on crushed Tetramin Tropical Tablets for Bottom Feeders. Pupae were collected by hand and allowed to emerge as adults in 0.03 m³ screened cages.

*Plasmodium berghei* ANKA parasites were a gift from the Johns Hopkins Malaria Research Institute and were maintained by infecting outbred Swiss Webster female mice (Charles River Laboratories). Frozen *P. berghei* infected mouse blood of ca. 25–40% parasitemia was diluted 1:1 with RPMI 164 media and 200 μl was injected interperitoneally into 2 mice. After 2 days, the level of parasitemia was measured by pricking the tail with a 26 G needle, smearing the blood onto a glass slide, fixing with 100% methanol for 10 s, and then staining with 10% Giemsa stain for 30 min. Slides were washed in deionized water and the smear measured to count the fraction of red blood cells that were infected with parasites ( = % parasitemia). When the blood in the infected mouse reached between 4 and 10% parasitemia, the mouse was killed and infected blood was immediately removed by heart puncture. The blood was diluted to 2% parasitemia with RPMI containing 6 I.U. heparin per ml. Two hundred μl of this solution was injected into a new mouse as described above. Thirty-six hours later, mosquitoes were allowed to feed on this mouse for 6 min.

Parasites were passed through mosquitoes to recover P0 blood (a parasite infection caused by the bite of an infected mosquito, rather than mouse-mouse transmission). In these cases, mosquitoes were fed for 30 min on the infected mouse and were allowed to rest at 19 °C for 25 days. On day 25 post infection, the *P. berghei* infected mosquitoes were allowed to feed on an uninfected mouse for 30–45 min. Seven days later, the *P. berghei* infection level in the mouse was measured and the mouse was killed and blood removed by heart puncture when the infection level reached 25–40%. Infected blood was mixed 1:1 with 30% (vol/vol) glycerol in PBS and 6 I.U heparin per ml. This mixture was then flash frozen in liquid nitrogen and stored in liquid nitrogen.

**Fluorescence analysis of *Asaia* strains**. Constitutive and BMI-driven GFP strains of *Asaia* were assessed for conditional fluorescence using both solid and liquid media. All media used in the remainder of the study was Davis Minimal media[57] supplemented with 5% (wt/vol) nectar solution made with a 1:2:1 ratio of sucrose:fructose:glucose in DI water. First, the putative BMI strains were streaked to single colonies on blood-supplemented (chocolate agar) and replete minimal agar to assess qualitative fluorescence. All human blood used in this study was obtained from a healthy, unpaid volunteer after informed consent was obtained by a licensed and certified physician assistant. The venipuncture was performed using sterile

technique and universal precautions. Any strains that showed an increase in fluorescence on the chocolate agar plates were analyzed in liquid culture. Strains were grown to log phase and separated into minimal broth with or without the addition of 2.5% (vol/vol) human blood. They were allowed to grow in these conditions for one h and optical density and fluorescence were measured using a SPECTRAMAX i3x from Molecular Devices with SoftMax Pro 7 software. The data were normalized to the fluorescence of a premeasured culture of the constitutively fluorescent *Asaia* strain ANB50 that had been separated into the varying media directly before analysis, to account for any signal quenching that may have resulted from the addition of the blood. Data was visualized in RStudio using barplot2 and dotplot[58]

In vivo activity of the promoters was assessed by observing the difference in fluorescence between blood and sugar-fed mosquitoes. The BMI strains, AGLR1 negative control, and ANB50 constitutive positive control were fed to mosquitoes at 0.1 $OD_{600}$ in a sugar meal. Bacteria were allowed to colonize the mosquito for 24 h; then half were fed blood and half sugar. The mosquitoes that fed on blood did so for 10 min on an anesthetized mouse. All mosquitoes were kept for 2 days at 29 °C and fed on sugar to allow time for the blood bolus to partially digest, after which the midguts of all mosquitos were dissected, washed once in PBS, and analyzed with fluorescent microscopy without other special preparations. Fluorescence in midguts was evaluated by eye.

**Assessing the conditionality of BMI scorpine strains**. To ensure that the effector scorpine was conditionally expressed from the antiplasmodial plasmids, the secretion of scorpine was evaluated from *Asaia* colonies grown on minimal media and chocolate agar through western blotting analysis. Each construct was grown to log phase in minimal media and verified through PCR before being streaked on both minimal media and chocolate agar plates. Control plates without bacteria were used to evaluate the amount of protein collected from the media itself. These plates were allowed to grow for 2 days at 30 °C before colonies were collected by flooding the plates with 1.5 ml of minimal media and gently scraping the cells from the plates. One ml of each cell suspension was collected and spun down at 13,800×g for 5 min. The supernatant from these cultures was collected and placed on ice while the pellet was resuspended in 20% (vol/vol) B-per (Thermo-Fisher) in TBS. The total protein for each of the pelleted samples was analyzed through a Bradford assay after accounting for the excess protein collected from the medium containing blood. Each supernatant sample was then diluted to the same total protein concentration based on the protein content of the pellets and 75 µl of each was added to 25 µl of 3x Laemmli buffer and boiled for 10 min. Fifteen µl of these samples were loaded into a 10% Mini-PROTEAN® TGX™ precast gel (BioRad) with a Precision Plus Protein™ Kaleidoscope™ size standard (BioRad). These were run at 200 V for 30 min after which they were transferred to a PVDF membrane in a BioRad transfer apparatus using Tris-Glycine transfer buffer (25 mM Tris, 150 mM glycine, 10% methanol) at 100 V for 60 min. The membrane was blocked using 4% BSA TBS-T solution (50 mM Tris-Cl, 150 mM NaCl, 0.1% Tween20, 4% (wt/vol) fraction V BSA, pH 7.5) overnight at 4 °C. The following day the blot was incubated with the bacterial alkaline phosphatase antibody (HRP) (GeneTex, cat. #GTX27319) at a 1:5,000 dilution suspended in blocking buffer and again incubated overnight at 4 °C. The membrane was then washed in cold TBS-T for 15 min four times. WesternBright Sirius chemiluminescent HRP substrate (Advansta, cat. # GTX27319) was applied to the membrane for 2 min in the dark before being visualized on an Odyssey Fc dual mode imaging system from LI-COR The uncropped western blot is shown in Supplementary Fig. 2.

**Fitness assessments of antiplasmodial *Asaia* strains**. Fitness of the antiplasmodial *Asaia* strains was tested in three ways. The first was by measuring the maximum growth rate of each strain. To do this, each *Asaia* was inoculated at 0.1 $OD_{600}$ in 200 µl of a 96 well plate. The $OD_{600}$ was analyzed over 24 h at 15 min intervals using a SPECTRAMAX i3x (Molecular Devices). The SoftMax Pro 7 software was used to create growth curves of collated replicates of each strain until they reached stationary phase. Growth curves were further analyzed using the package growthrates[59] in RStudio to find the maximum growth rate of each strain of *Asaia*. Data was visualized in RStudio using boxplot[58]

The ability of the transgenic strains to survive when competed in a culture with wild-type *Asaia* was tested in a competition experiment. At least ten replicates of each transgenic strain were grown to log phase (ACG18, n = 10; ACG18.Hem, n = 11; ACG18.HF, n = 12; ACG18.HlyA, n = 10; ACG18.HlyC, n = 12) and mixed with wild-type *Asaia* in a 50/50 ratio of a 0.5 $OD_{600}$ culture. These mixed cultures were allowed to grow for 6 h, then diluted to $1.0 \times 10^{-5}$ $OD_{600}$ of which 100 µl was plated on minimal media with or without 120 µg ml$^{-1}$ kanamycin. The ratio of transgenic bacteria to wild-type was calculated by comparing the CFUs on the selective media to the CFUs on the nonselective media. Data was visualized in RStudio using boxplot[58]

The final fitness assessment performed was a mosquito colonization experiment that was repeated in triplicate. Each strain was fed to female *An. stephensi* mosquitoes at a 0.1 $OD_{600}$ dilution in the sugar meal. After 48 h, mosquito midguts were dissected and homogenized using a tissue grinder. Ten midguts for each strain were pooled and diluted to 100 µl PBS per midgut. These samples were again diluted 100-fold in PBS and 100 µl of each dilution was plated on kanamycin

supplemented minimal media. CFUs for each strain were counted and compared to the total number of CFUs collected across test groups. Data was visualized in RStudio using boxplot[58]

**Plasmodium berghei inhibition analysis**. To evaluate the inhibitory effect of the *Asaia* strains containing the antiplasmodial plasmid constructs, adult female ND4 Swiss Webster mice were infected with *Plasmodium berghei* ANKA[60]. Parasites were allowed to develop in the mice until parasitemia level reached 4–10%. At this point the mice were killed and blood was collected via cardiac puncture. The infected blood was diluted with RPMI media to 2% parasitemia, then 200 µl (5 × $10^7$ parasites) was injected intraperitoneally into an uninfected mouse. At the time of this transfer, each *Asaia* strain to be tested was fed to *An. stephensi* mosquitoes at a 0.1 $OD_{600}$ dilution in the sugar meal. Thirty-six hours post-infection each test group of mosquitoes was blood-fed on the infected mouse for 6 min each. The order of the test groups was rotated randomly and blindly for each of the 3 trials. The ability of the parasite to undergo exflagellation was also tested at this time using 6 µl ookinete media (1 L RPMI media supplemented with 2 g sodium bicarbonate, 50 mg hypoxanthine, 20.5 mg xanthurenic acid) mixed with 10% fetal bovine serum, 2 µl of 1 mg ml$^{-1}$ of heparin in PBS, and 2 µl of blood collected from a tail prick of the mouse. At least 2 exflagellation events occurred for each malarial trial. Exflaggellation occurs when microgametes exit red blood cells after a female mosquito takes a *Plasmodium*-infected blood meal and can be monitored by microscopy. The number of these events in the blood meal is a measure of how infectious it is to the mosquito.

Parasites were allowed to develop in the mosquitoes for 12 days at 19 °C in order to form oocysts. After 12 days, the mosquito midguts were dissected and stained with a 100-fold dilution of mercurochrome stain for 2 min. They were then left to destain for 5 min in PBS. The midguts were analyzed at 100x magnification and the number of oocysts per midgut were counted for each test group. All steps in this process were performed blindly and ordered randomly. Data was visualized in RStudio using Bee Swarm.[61]

**Statistics and reproducibility**. Significance was set to $P < 0.05$. Variance was estimated using standard error of the mean and is appropriately similar between test groups of each experiment. Significance of the mean was calculated using Welch's two-tailed t-test (Fig. 1 and Supplementary Fig. 1) or one-way ANOVA with Dunnett's correction (Fig. 5) in RStudio appropriate for multiple comparisons to a single control with normal distribution unless otherwise noted.

The qualitative evaluation of fluorescent midguts represented in Fig. 2 was repeated at least 3 times per strain, and over 5 times for the control groups as well as AGLR1.Hem. Each trial consisted of at least 20 midguts per strain, therefore, 10 midguts per test group.

In Fig. 6, the data are pooled from 3 individual experiments. The median value of oocycts per midgut was calculated by and compared between treatments using quantile regression[62] in RStudio. Quantile regression is a non-parametric test that compares subsets of a data set individually and is useful for data showing unequal variation[62]. The significance of the difference in *P. berghei* oocyst prevalence was evaluated using binomial $\chi^2$ tests with 1° of freedom. All colony and oocyst counts were done blindly regarding which strain and condition was evaluated, and the strains were ordered randomly.

**Code availability**. All data generated in R Studio (version 1.0.136) was obtained using standard coding for the programs listed for each experiment. This code is available upon request and no custom code was created. The actual code used in this study was deposited at Dryad Digital Repository doi:10.5061/dryad.s85j284.

## Data availability

All experimentally generated data are readily available upon request of the corresponding author. In addition, quantitative data and R-based code used in Figs. 1, 5a-c, and 6 were deposited at Dryad Digital Repository doi:10.5061/dryad.s85j284. The original, uncropped western blot from Fig. 4 is provided in Supplementary Fig. 2. Sequences of plasmids generated in this study have been deposited in GenBank under the accession codes MG702576 though MG702589, and MH734181 for pCG6.

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

## Acknowledgements

*Asaia sp.SF2.1* was a gift of G. Favia of the University of Camerino, Italy. *An. stephensi and P. bergei ANKA* were gifts of M. Jacobs-Lorena and the John Hopkins Malaria Research Institute. We thank R. Shanks for his critical review of the manuscript and B. Calhoun for expert technical assistance. Research reported in this publication was supported entirely by the National Institute of Allergy and Infectious Diseases of the National Institutes of Health under award number R15 AI107735. The content is solely the responsibility of the authors and does not necessarily represent the official views of the National Institutes of Health.

## Author contributions

J.L.S and D.J.L. conceived the research and prepared the manuscript. J.L.S. and C.L.G. conducted the experiments. C.C. conducted the growth curve experiments.

## Additional information

**Competing interests:** The authors declare no competing interests.

