## [Peer Review File · Nature Communications]

Reviewers' Comments:

Reviewer #1:

Remarks to the Author:

Genetic manipulation of the mosquito microbiota, or paratransgenesis, is regarded with high interest in the context of developing malaria transmission-blocking strategies. This study presents a novel paratransgenesis approach, where a bacterial strain from the microbiota, *Asaia* sp. SF2.1, is manipulated to produce an antimicrobial peptide with antimalarial properties under inducible conditions. This is a very interesting idea, as the production of an antimicrobial peptide is likely to reduce the fitness of the engineered bacterial strain. Overall, this manuscript shows very interesting data and also suffers important drawbacks.

The strengths of this study are that the authors found some promoters for bloodmeal-inducible expression, they managed to produce the antimicrobial peptide under the control of these promoters and showed a strong reduction in infection by murine malaria parasites *P. berghei*. However, the manuscript still lacks some work to make strengthen the authors' claims. The fitness cost of scorpine on *Asaia* is not monitored. If there is no fitness cost, then there is no claim for the use of such inducible promoters. A fitness cost on other bacterial strains of the microbiota may even compromise a homeostatic microbiota. If there is a fitness cost, this makes the claim of the inducible promoters strong, but the authors should verify (1) that the expression of scoprin is indeed induced by the bloodmeal and (2) that scorpine-producing bacteria are kept over several generations while keeping their potential to reduce *Plasmodium* infection. Infections with human malaria parasites would also be a plus.

The manuscript itself needs a lot of improvement. The figures can be restructured so that several panels are grouped by project aim. The experiments need further explanations in the results. Conventional use of the *, ** and *** should be respected when mentioning statistical results. Figure legends should be homogenized to contain statistics, number of replicates, and should be self-explanatory. Material and methods are incomplete, e.g. there is no mention on mosquito rearing and mosquito manipulation. Results and materials and methods are not always consistent on the methods (e.g. line 117 and line 301: 24h vs 36h exposure to the bacteria). There are also some many typos that they cannot be listed here. The last four articles of the references are not mentioned in the text.

Figure 1 probably contains an error in the "Kan" and "No antibiotic" label.

Reviewer #2:

Remarks to the Author:

I have enjoyed reading the manuscript entitled "Blood meal-induced inhibition of vector-borne disease by transgenic microbiota". The subject is of interest for a wide audience (people working with vector borne diseases) and contains also technical tips that could be of interest for an even wider audience.

The research plan is clear, the methods implied coherent with the objectives and results original and able to open new perspectives in the field of vector borne diseases control.

M&M are nicely reported likely allowing other scientists to replicate the experiments. Figures and tables well represents the experiments and the results achieved.

In my view this manuscript deserves publication in nature communications.

Nevertheless, I have a couple of concerns that should be addressed before final acceptance:

- 1) Lines 257-258: Are the authors sure that the amount of antibiotics were 30 and 120 milligrams/milliliter? I suppose were nanogram/milliliters.
- 2) Bibliography: This section needs to be totally revised. Too many mistakes, too many failures to comply with the journal's instructions!

Reviewer #3:

Remarks to the Author:

Previous studies have shown that bacteria residing in mosquito guts can be used in paratransgenic approaches to alter the capacity these vectors to transmit malaria. Specifically, these past studies have focused on establishing the system in *Anopheles* mosquitoes to reduce *Plasmodium* infection and examining the heritability of the symbiotic bacteria, but no studies, to my knowledge, have focuses on alleviating the fitness costs of expressing a transgene expression in the bacterium. Here, Shane and co-workers describe the identification and exploitation of blood meal inducible promoters to drive anti-malarial effector proteins in *Asaia*. Importantly, these transgenic lines of *Asaia* have reduced fitness costs, both in vitro but more importantly, in vivo. Furthermore, these bacteria strains that use a blood meal inducible promoter to drive the anti-malarial effector transgene significantly inhibit *Plasmodium* compared to the constitutive expression of the same effector. This paper is an important contribution to the malaria paratransgenesis field and substantially advances this discipline in terms of developing a system that could be appropriate for use in the field. I have a few comments and points of concern below but these should be easily addressable.

Notably, scorpine driven by these blood meal inducible promoters has a greater efficacy in pathogen suppression in terms of prevalence. Quite appropriately the authors highlight this important finding, which likely has greater epidemiological ramifications than reducing parasite load. However, one point the authors failed to address in the discussion is the difference in parasite load of artificial lab infections compared to the burden of *Plasmodium* in field mosquitoes. These lab experiments have an artificially high parasite burden. Field oocyst infection rates range from 1-2 parasites/midgut. As such, this strategy would likely be even more effective in terms of reducing prevalence when comparing parasite burdens under a field appropriate parasite density.

These experiments were conducted in lab-reared mosquitoes which have a different microbiome compared to their field counterparts, most notably in the alpha diversity, or number of different bacterial species present in the mosquito. As such, in the field, the introduced transgenic *Asaia* will likely encounter many more species of bacteria in the guts of mosquitoes compared to a lab setting. This point could be briefly mentioned in the discussion.

L53. The authors might want to clarify that *Wolbachia* approaches are not paratransgenic.

L75. What is meant by parasitic resistance? The natural anti-pathogen actively of the *Asaia*? Has this been shown?

Stylistic comment – perhaps figures 5 and 6 could be incorporated together as a panel figure?

Also, it would be helpful if the colour code in the figures remained constant throughout the figures. (Fig 5/6 vs 7).

Minor comments:

L10. The phrase easily transmissible seems a bit awkward.

L10 – refractory to the pathogen (not the disease). And further in the MS (Line 173/L179). Mosquitoes transmit pathogens, not disease.

Figure 2. It might be helpful to say these images are representative of all the samples observed. The samples size is impressive.

L73. Large? Perhaps could be rephrased. Depends on your definition of large.

There are several more recent paratransgenesis reviews that might be worth citing.

Grant Hughes

Reviewer #4:

Remarks to the Author:

The naturally occurring midgut symbiont, *Asaia* sp. SF2.1, was genetically engineered to conditionally express a known antimalarial effector gene, *scorpine*. Conditional expression was achieved by isolating orthologous promoters within *Asaia* that have been demonstrated to be activated upon encountering blood meal-like conditions with other bacterial species. Importantly, this conditional expression allows the transgenic bacteria to compete effectively with native *Asaia* within mosquito midgut. This *Asaia* bacterium is also found in several other significant vectors such as *Ae. aegypti* and *Ae. albopictus* as well as *Scaphoideus titanus*, the vector of grapevine phytoplasma, so it may have a wide range of applicability towards different disease settings, and be of interest to others in the community and the wider field.

Major comments:

1. Line 25; "Most of the disease causing pathogens are transmitted through blood ingestion by arthropod vectors" is not true, many other more substantial transmission modes for infectious diseases...think airborne, fecal-oral, sexual, etc.
2. While it is true that *Serratia marcescans* may be an opportunistic pathogen for humans, how closely related is the *Serratia* AS1? For example, most *E. coli* are not pathogenic, and O157:H7 is an exception rather than the rule. In the same vein, *Asaia* could become pathogenic, like any other bacterium, if picking up certain genetic loci/and or developing resistance, or landing in a new environmental niche (i.e. hospitals with access to opportunistic infections due to immunocompromised population).
3. Citation provided for *Asaia* within *Anopheles* mosquitoes is for *A. stephensi*, please provide citations for *Asaia* in other *Anopheles* species?
4. Multiple references are incomplete and/or incorrect.
5. Ethical consideration statement should be included due to the use of vertebrate (human) blood.
6. Line 291; What is the source of the human blood?
7. What was used to measure the *in vivo* activity of promoters within mosquitoes? More specifics needed on the fluorescent microscopy and how the mosquito midguts were prepared.
8. Methods; Mosquito identification required...and how were the mosquitoes reared?
9. Lines 333 and 348; *Anopheles stevensi*? I think that you mean *An. stephensi*?
10. The fitness assessment within mosquitoes is weak, and not really a fitness assay, what is actually determined is survival after 36 hours within mosquitoes. Further, these mosquitoes are sugar fed. What is the impact of blood feeding towards growth and persistence? Also fitness in the true sense, would be about evolutionary persistence, thus, how many of the transgenics were vertically transmitted?
11. For the non-Plasmodium readers, what is the purpose of testing for the ability of the parasite to undergo exflagellation?
12. Figure 1: what is the purpose of antibiotic versus no antibiotic, is this representing fluorescence induction within the chocolate agar versus the minimal media without blood? This figure needs better clarification within the legend.
13. Lines 116-117; The bacteria were allowed to colonize the mosquito for 24h is different than what is described in the materials and methods of 36 hours.
14. In Figure 4, were growth rates also statistically analyzed between BMI isolates and SF2.1?
15. For the results of Figure 6, could there be a colonization benefit that is provided by the WT *Asaia* that is already in these mosquitoes?
16. Figure 7 should also include the quantitative abundance of transgenic *Asaia* at the point of oocyst development (i.e. 12 days).

17. Was there any verification that scorpine is indeed produced by the transgenic Asaia under BMI conditions? This evidence is important.

18. In the discussion, the results should be put in broader context. Right now, it is mostly rehashing the results.

19. Lines 221-223; Is there precedence for stating that the few remaining oocysts may not have produced many functional sporozoites? This could be examined.

Minor comments:

1. Provide a citation for Line 22-23

2. Line 27; replace treatment with measures for these diseases...these vector control strategies are not "treatments"

3. Lines 31-32; "changing the vectors phenotypically" is confusing, please clarify.

4. Line 55; compromised

5. Line 77; One of the most lethal vector-borne diseases is

6. Figure 4 legend; Growth curves and maximum growth rates were calculated in RStudio.

Response to Reviewer's Comments

We thank the reviewers for their many useful comments. Incorporating them significantly clarified and strengthened the revised manuscript. All changes from the previous version of the manuscript are noted in yellow in the new version. Below is a response to the specific concerns of each reviewer. We have repeated the reviewer's comments verbatim in italics.

Reviewers' comments:

Reviewer #1 (Remarks to the Author):

*Genetic manipulation of the mosquito microbiota, or paratransgenesis, is regarded with high interest in the context of developing malaria transmission-blocking strategies. This study presents a novel paratransgenesis approach, where a bacterial strain from the microbiota, *Asaia* sp. SF2.1, is manipulated to produce an antimicrobial peptide with antimalarial properties under inducible conditions. This is a very interesting idea, as the production of an antimicrobial peptide is likely to reduce the fitness of the engineered bacterial strain. Overall, this manuscript shows very interesting data and also suffers important drawbacks.*

*The strengths of this study are that the authors found some promoters for bloodmeal-inducible expression, they managed to produce the antimicrobial peptide under the control of these promoters and showed a strong reduction in infection by murine malaria parasites *P. berghei*. However, the manuscript still lacks some work to make strengthen the authors' claims. The fitness cost of scorpine on *Asaia* is not monitored. If there is no fitness cost, then there is no claim for the use of such inducible promoters. A fitness cost on other bacterial strains of the microbiota may even compromise a homeostatic microbiota. If there is a fitness cost, this makes the claim of the inducible promoters strong, but the authors should verify (1) that the expression of scorpine (sic) is indeed induced by the bloodmeal and (2) that scorpine-producing bacteria are kept over several generations while keeping their potential to reduce *Plasmodium* infection.*

Response: It is not at all clear to us what the reviewer means by "*The fitness cost of scorpine on *Asaia* is not monitored*". The fitness cost to producing scorpine comes either from the negative cost of the peptide itself acting on *Asaia*, the cost of producing it constitutively, or both. We did not try to disentangle these possibilities. Constitutive expression of heterologous genes in bacteria often causes fitness effects, which is why so much time and effort has been spent (in *E. coli*, for example) to work around these problems (this latter point is highlighted in lines 205-210 in the new ms). Based on what we know about *E. coli*, we hypothesized that placing the transcription of scorpine under the conditional control of a blood-meal induced promoter would improve its fitness in non-blood meal conditions leading to improved paratransgenesis. The fact that the *Asaia* strain that constitutively expresses scorpine is less fit than wild type is easy to measure and not under dispute. Fitness effects of the constitutive expression of scorpine are obvious and significant when we measured growth rate, ability to compete with wild-type *Asaia*, and ability to colonize the mosquito midgut (new figure 4). In every case, ACG18 (the constitutive strain) performed very poorly compared to wild-type *Asaia*.

Putting the scorpine gene under conditional control significantly improved the performance of the strains (with most promoters in most measurements). The fitness cost of constitutive expression of scorpine is obvious. It is irrelevant whether the fitness cost comes from the peptide itself. The use of conditional promoters vastly improves the performance of the paratransgenic strains. That is the point of the manuscript.

We ensured that scorpine was being expressed in a conditional fashion (like GFP) using western analysis. This is shown in Supplementary Figure 1 and described in the supplementary methods. We chose to use a comparison on agar plates with and without blood since it allowed us to better control the amount of bacteria used in the experiment, something that is very difficult using blood-fed mosquitoes. The results are very clear: scorpine is only being made in the presence of blood in BMI strains while it is always produced in the constitutive strain.

We agree with the reviewer that the expression of scorpine may have an effect on other species in the microbiota. The proper way to test this would be to perform a next generation sequencing experiment on the mosquito microbiota 1) before we introduce Asaia, 2) after we introduce wild-type Asaia, and 3) in the presence of paratransgenic Asaia. That is a major undertaking that, while interesting, is far outside the scope of this work and, moreover, is not immediately relevant to the idea of improving paratransgenesis by placing antiplasmodial genes under conditional control.

We agree with the reviewer that improved strain fitness should lead to increased ability to be transmitted across mosquito generations. Unfortunately, an unambiguous experiment of that kind is not possible using the strains we developed here. Our strains are built using plasmids that require drug selection to be maintained for long periods of time (like the weeks it would take for a multigenerational mosquito experiment). If we were to perform an intergenerational experiment using these strains and used kanamycin to maintain them, the fitness advantage afforded by the drug selection would almost certainly swamp any fitness advantage provided by conditional expression of scorpine, or would at least be difficult to untangle from that conferred by drug resistance. Similarly, if we performed the experiment without drug selection, the plasmids would be lost making any fitness gain from conditional expression difficult to interpret. The proper way to determine if conditional expression of scorpine leads to increased persistence over several mosquito generations would be to create strains that carry these constructs on the chromosome, at the same site in the chromosome to avoid position effects. That technology does not exist for Asaia at this time, although we are working to develop it.

Infections with human malaria parasites would also be a plus.

Response: Our institution does not have a facility suitable for housing *P. falciparum*-infected mosquitoes safely, so these experiments were not performed. Moreover, where paratransgenic bacteria have been evaluated against both *P. berghei* and *P. falciparum* (e.g., with *P. agglomerans*), the paratransgenic effects of scorpine are consistent between the two parasite species, working arguably better against *P. falciparum*. We use *P. berghei* because it is safe for us to do so under our institutional constraints.

The manuscript itself needs a lot of improvement. The figures can be restructured so that several panels are grouped by project aim.

Response: We grouped all the fitness experiments together in a new figure 4 (combining old figures 4, 5, and 6).

*The experiments need further explanations in the results. Conventional use of the *, ** and *** should be respected when mentioning statistical results. Figure legends should be homogenized to contain statistics, number of replicates, and should be self-explanatory.*

Response: We standardized the use of *, **, and *** in all figures setting them to values of $P < 0.05$, 0.01 , and 0.001 , respectively. All figure legends (where statistics are used, namely, Figs 1, 4 and 5) contain information on sample size or else those values are in the figures themselves (i.e., fig 5). We also improved the section in the methods that describe the statistical analyses (lines 392-407).

Material and methods are incomplete, e.g. there is no mention on mosquito rearing and mosquito manipulation.

Response: Mosquito rearing and parasite manipulation are described in the methods, lines 265-296.

Results and materials and methods are not always consistent on the methods (e.g. line 117 and line 301: 24h vs 36h exposure to the bacteria).

Response: These values are corrected and are consistent (lines 116 and 328).

There are also some (sic) many typos that they cannot be listed here.

Response: As far as we know, there are no remaining typos.

The last four articles of the references are not mentioned in the text.

Response: These references referred to the supplementary information. They have been moved to that section.

Figure 1 probably contains an error in the “Kan” and “No antibiotic” label.

Response: The reviewer is correct that there was an error in the labeling of this figure. Those labels have been removed and should have read “Blood” and “Minimal media”.

Reviewer #2 (Remarks to the Author):

I have enjoyed reading the manuscript entitled "Blood meal-induced inhibition of vector-borne disease by transgenic microbiota". The subject is of interest for a wide audience (people working with vector borne diseases) and contains also technical tips that could be of interest for an even wider audience.

The research plan is clear, the methods implied coherent (sic) with the objectives and results original and able to open new perspectives in the field of vector borne diseases control. M&M are nicely reported likely allowing other scientists to replicate the experiments. Figures and tables well represents the experiments and the results achieved.

In my view this manuscript deserves publication in nature communications.

Nevertheless, I have a couple of concerns that should be addressed before final acceptance:

1) Lines 257-258: Are the authors (sic) sure that the amount of antibiotics were 30 and 120 milligrams/milliliter? I suppose were nanogram/milliliters.

Response: The reviewer is correct that the concentrations were wrong. These have been corrected (lines 249-250).

2) Bibliography: This section needs to be totally revised. Too many mistakes, too many failures to comply with the journal's instructions!

Response: The references have been checked and corrected (lines 415-549).

Reviewer #3 (Remarks to the Author):

Previous studies have shown that bacteria residing in mosquito guts can be used in paratransgenic approaches to alter the capacity these vectors to transmit malaria. Specifically, these past studies have focused on establishing the system in Anopheles mosquitoes to reduce Plasmodium infection and examining the heritability of the symbiotic bacteria, but no studies, to my knowledge, have focuses on alleviating the fitness costs of expressing a transgene expression in the bacterium. Here, Shane and co-workers describe the identification and exploitation of blood meal inducible promoters to drive anti-malarial effector proteins in Asaia. Importantly, these transgenic lines of Asaia have reduced fitness costs, both in vitro but more importantly, in vivo. Furthermore, these bacteria strains that use a blood meal inducible promoter to drive the anti-malarial effector transgene significantly inhibit Plasmodium compared to the constitutive expression of the same effector. This paper is an important contribution to the malaria paratransgenesis field and substantially advances this discipline in terms of developing a system that could be appropriate for use in the field. I have a few comments and points of concern below but these should be easily addressable.

Notably, scorpine driven by these blood meal inducible promoters has a greater efficacy in pathogen suppression in terms of prevalence. Quite appropriately the authors highlight this important finding, which likely has greater epidemiological ramifications than reducing parasite load. However, one point the authors failed to address in the discussion is the difference in parasite load of artificial lab infections compared to the burden of Plasmodium in field mosquitoes. These lab experiments have an artificially high parasite burden. Field oocyst infection rates range from 1-2 parasites/midgut. As such, this strategy would likely be even more effective in terms of reducing prevalence when comparing parasite burdens under a field appropriate parasite density.

Response: The reviewer is correct about the difference between parasite load in the field vs in the lab, especially with *P. berghei* which is notorious for producing very high infection rates. We spent a lot of time working out conditions that keep our infections (relatively) low, but they still are much higher than those encountered in the field. We pointed this out in the discussion (lines 234-238).

These experiments were conducted in lab-reared mosquitoes which have a different microbiome compared to their field counterparts, most notably in the alpha diversity, or number of different bacterial species present in the mosquito. As such, in the field, the introduced transgenic Asaia will likely encounter many more species of bacteria in the guts of mosquitoes compared to a lab setting. This point could be briefly mentioned in the discussion.

Response: We included this point in the new ms (lines 210-213).

L53. The authors might want to clarify that *Wolbachia* approaches are not paratransgenic.

Response: We noted that using *Wolbachia* is a “similar” approach to paratransgenesis (line 55). *Wolbachia* is, in fact, used to alter the phenotype of the mosquito populations into which it is introduced, either producing a life-span shortening phenomenon or refractoriness towards viral infection. That certainly seems similar to paratransgenesis.

L75. What is meant by parasitic resistance? The natural anti-pathogen activity of the Asaia? Has this been shown?

Response: Asaia does not have any natural anti-pathogen activity. We changed “parasitic resistance” to “pathogen inhibition engineered in....” in the new ms (line 76).

Stylistic comment – perhaps figures 5 and 6 could be incorporated together as a panel figure?

Response: We combined figures 5, 6, and 7 into a new figure 4.

Also, it would be helpful if the colour code in the figures remained constant throughout the figures. (Fig 5/6 vs 7).

Response: All color coding is consistent between figures where needed, especially figures 4 and 5.

Minor comments:

L10. The phrase easily transmissible seems a bit awkward.

Response: Agreed. We removed the phrase.

L10 – refractory to the pathogen (not the disease). And further in the MS (Line 173/L179). Mosquitoes transmit pathogens, not disease.

Response: Again, agreed. We combed the entire manuscript for this kind of confusion. We replaced every instance of “antimalarial” with “antiplasmodial” to reflect this point. Many thanks for catching this.

Figure 2. It might be helpful to say these images are representative of all the samples observed. The samples size is impressive.

Response: We included such phraseology in the figure legend for Figure 2 (line 583).

L73. Large? Perhaps could be rephrased. Depends on your definition of large.

Response: We changed “large” to “variety” (line 74).

There are several more recent paratransgenesis reviews that might be worth citing.

Response: We included two reviews originally (references 4 and 9) but chose not to include any additional ones in the new ms, in part because we are already exceeding the maximum reference number as outlined in the Nature instructions (<https://www.nature.com/nature/for-authors/formatting-guide>)! We feel like the reviews we cite, plus the primary literature, does an adequate job in introducing the reader to genetic and paratransgenic modification of insects.

Reviewer #4 (Remarks to the Author):

The naturally occurring midgut symbiont, Asaia sp. SF2.1, was genetically engineered to conditionally express a known antimalarial effector gene, scorpine. Conditional expression was achieved by isolating orthologous promoters within Asaia that have been demonstrated to be activated upon encountering blood meal-like conditions with other bacterial species. Importantly, this conditional expression allows the transgenic bacteria to compete effectively with native Asaia within mosquito midgut. This Asaia bacterium is also found in several other significant vectors such as Ae. aegypti and Ae. albopictus as well as Scaphoideus titanus, the vector of grapevine phytoplasma, so it may have a wide range of applicability towards different disease settings, and be of interest to others in the community and the wider field.

Major comments:

1. Line 25; “Most of the disease causing pathogens are transmitted through blood ingestion by arthropod vectors” is not true, many other more substantial transmission modes for infectious diseases...think airborne, fecal-oral, sexual, etc.

Response: Agreed. We changed “most” to “many” (line 26).

2. While it is true that *Serratia marcescans* may be an opportunistic pathogen for humans, how closely related is the *Serratia AS1*? For example, most *E. coli* are not pathogenic, and O157:H7 is an exception rather than the rule. In the same vein, *Asaia* could become pathogenic, like any other bacterium, if picking up certain genetic loci/and or developing resistance, or landing in a new environmental niche (i.e. hospitals with access to opportunistic infections due to immunocompromised population).

Response: The comments about strains of *Serratia marcescens* being major opportunistic pathogens is a matter of the scientific record, just as it is true that *Asaia* is not. As far as we know, the people working on developing *Serratia* as paratransgenesis platforms have not evaluated their stains for human pathogenicity. We feel that one of the most important choices that can be made regarding paratransgenesis is a careful choice of bacterial species with which to work. We have a freezer full of different bacterial species isolated from mosquitoes, some of which colonize mosquitoes readily and are passed intergenerationally but that have strains that are major opportunistic human pathogens. We have not pursued these intentionally. It seems foolish to invest heavily in a platform that may prove to be pathogenic or that regulators will heavily scrutinize (and perhaps reasonably reject) later. We feel that the lack of human pathogenicity shown by *Asaia* in the literature is a major point in its favor and is worth pointing out.

3. Citation provided for *Asaia* within *Anopheles* mosquitoes is for *A. stephensi*, please provide citations for *Asaia* in other *Anopheles* species?

Response: A reference has been added demonstrating the presence of *Asaia* in *An. gambiae* (ref #19).

4. Multiple references are incomplete and/or incorrect.

Response: As far as we know, the references are complete, correct, and formatted properly.

5. Ethical consideration statement should be included due to the use of vertebrate (human) blood.

and

6. Line 291; What is the source of the human blood?

Response: An ethical statement and the source of the blood are included in the methods (lines 313-315).

7. What was used to measure the in vivo activity of promoters within mosquitoes? More specifics needed on the fluorescent microscopy and how the mosquito midguts were prepared.

Response: Promoter activity driving GFP in blood fed mosquito midguts (figure 2) was evaluated by eye. The activity was not quantified by any other assay. This is noted in the figure legend and in the methods (lines 579 and 333-334, respectively). We were not looking for subtle effects and the strains we pursued were obviously either glowing or not, just as the figures show. Midgut prep is noted in the methods (lines 333-334).

8. Methods; Mosquito identification required....and how were the mosquitoes reared?

Response: Mosquito rearing conditions and the criteria for their identification are included in lines 265-274.

9. Lines 333 and 348; Anopheles stevensi? I think that you mean An. stephensi?

Response: Yes, we meant stephensi! This is correct throughout the ms.

10. The fitness assessment within mosquitoes is weak, and not really a fitness assay, what is actually determined is survival after 36 hours within mosquitoes. Further, these mosquitoes are sugar fed. What is the impact of blood feeding towards growth and persistence? Also fitness in the true sense, would be about evolutionary persistence, thus, how many of the transgenics were vertically transmitted?

Response: The fitness measurements that we made are not “weak” but are, in fact, standard ones used in the microbiology community. Fitness is defined, of course, in terms of survival and reproduction, thus survival in a mosquito midgut under sugar conditions is entirely germane.

The entire point of this manuscript was to show that placing an antiplasmodial effector gene under conditional regulation would lead to a superior paratransgenesis outcome, which it did. This outcome correlates well with the increased fitness that we can measure by standard means. The increased paratransgenic effect is most likely due to the increased ability to colonize the mosquito midgut before a blood meal (i.e., under sugar-fed conditions), which we measured. That means that more bacteria are present in the midgut at the time of the infectious blood meal. Whether or not this translates to increased transmission from mosquito generation to mosquito generation is an entirely separate question, although it can, in principle, be tested, but not with the strains we developed here. Our strains are built using plasmids that require drug selection to be maintained for long periods of time (like the weeks it would take for a multigenerational mosquito experiment). If we were to perform an intergenerational experiment using these strains and used kanamycin to maintain them, the fitness advantage afforded by the drug selection would almost certainly swamp any fitness advantage provided by conditional expression of scorpine, or would at least be difficult to untangle from that conferred by drug resistance. Similarly, if we performed the experiment without drug selection, the plasmids would be lost making any fitness gain from conditional expression difficult to interpret. The proper way to determine if conditional expression of scorpine leads to increased persistence over several mosquito generations would be to create strains that carry these constructs on the chromosome, at the same site in the chromosome to avoid position effects. That technology does not exist for *Asaia* at this time, although we are working to develop it.

11. For the non-Plasmodium readers, what is the purpose of testing for the ability of the parasite to undergo exflagellation?

Response: Exflagellation is explained in the methods (lines 383-385).

12. Figure 1: what is the purpose of antibiotic versus no antibiotic, is this representing fluorescence induction within the chocolate agar versus the minimal media without blood? This figure needs better clarification within the legend.

Response: Figure 1 was mislabeled. Those labels have been removed and should have read "Blood" and "Minimal media".

13. Lines 116-117; The bacteria were allowed to colonize the mosquito for 24h is different than what is described in the materials and methods of 36 hours.

Response: The correct timing is 24h and is the same in lines 116 and line 328.

14. In Figure 4, were growth rates also statistically analyzed between BMI isolates and SF2.1?

Response: Yes (now, figure 4a). This is stated in the text (line 139).

15. For the results of Figure 6, could there be a colonization benefit that is provided by the WT Asaia that is already in these mosquitoes?

Response: This is now figure 4c. Perhaps, but this effect would be present in all treatments and if present, is thus controlled for. The entire field of microbiota effects on paratransgenic strains (and vice versa) is an underexplored area. It is also out of the scope of this immediate study.

16. Figure 7 should also include the quantitative abundance of transgenic Asaia at the point of oocyst development (i.e. 12 days).

Response: Oocysts begin development after ookinete invasion of the midgut epithelium between 24-48 h after an infectious blood meal. Oocysts continue development over a course of about 2 weeks. It is not clear to us what the abundance of our strains at day 12 would mean. Any killing of parasites by paratransgenic *Asaia* is likely to occur before ookinete invasion within the midgut and thus before oocyst development. The oocysts that do manage to develop are due to parasites that escaped killing within the midgut. The abundance of *Asaia* on day 12 is irrelevant.

17. Was there any verification that scorpine is indeed produced by the transgenic Asaia under BMI conditions? This evidence is important.

Response: We ensured that scorpine was being expressed in a conditional fashion (like GFP) using western analysis. This is shown in Supplementary Figure 1 and described in the supplementary methods. We chose to use a comparison on agar plates with and without blood since it allowed us to better control the amount of bacteria used in the experiment, something that is very difficult using blood-fed mosquitoes. The results are very clear: scorpine is only being made in the presence of blood in BMI strains while it is always produced in the constitutive strain.

18. In the discussion, the results should be putter in broader context. Right now, it is mostly rehashing the results.

Response: We attempted to do this by clarifying the rationale of putting the scorpine gene under conditional control, especially see lines 204-216.

19. Lines 221-223; Is there precedence for stating that the few remaining oocysts may not have produced many functional sporozoites? This could be examined.

Response: There is one publication in the literature that we are aware of where oocysts were detected but **not** sporozoites. This study involves transgenic mosquitoes. This is Isaacs et al. (2012) Proc Natl Acad Sci U S A. Jul 10;109(28):E1922-30, table 3. Our reference to it was speculative in the original manuscript and we removed it from this version

Minor comments:

1. Provide a citation for Line 22-23

Response: Reference #1 was provided to document global malaria incidence (line 24).

2. Line 27; replace treatment with measures for these diseases...these vector control strategies are not "treatments"

Response: We changed "treatments" to "measures for" (line 28)

3. Lines 31-32; "changing the vectors phenotypically" is confusing, please clarify.

Response: We changed "changing the vectors phenotypically" to "changing the ability of vectors to transmit pathogens" (line 33).

4. Line 55; compromised

Response: Spelling changed (line 57)

5. Line 77; One of the most lethal vector-borne diseases is

Response: We changed this phrase to "One of the most serious vector-borne diseases is" (line 78).

6. Figure 4 legend; Growth curves and maximum growth rates were calculated in RStudio.

Response: The legend (now figure 4a) reads "Growth curves and maximum growth rates were calculated from 10 individual isolates of each strain grown over log phase of the bacteria using RStudio"

Reviewers' Comments:

Reviewer #1:

Remarks to the Author:

I am surprised about the patronizing tone used in the point-by-point response. My view of the review process is that reviewers are working for free to help authors improve a manuscript. Some phrasing may have been avoided in this context. However, this is just a comment and this review deals solely with the quality of the revision.

The research described in the previous version of the manuscript was good enough to be considered for revision at Nature Communications. After revision, there has been some improvement in the manuscript itself, even though there are still some small errors (e.g. line 137: Figure 4; Line 205: E. Coil; Line 300: BLAST is not a database; Line 395: AVOVA). In terms of experiments, except from a western blot on bacterial cultures, the authors did not follow any advice to improve their experimental design. For instance, although the vertical transmission issue has been raised by 2 reviewers, the authors decided not to perform them. Apparently, this point put light on an important caveat of this study, that bacteria are quickly lost by mosquitoes. Would it have been that much work to create a stable line expressing the transgene? At least, the issue could have been discussed in the revised manuscript. Considering scorpine expression in the blood-fed mosquito, normalizing to the number of bacteria (1) would not be difficult and (2) would not even be required as we just need to see whether scorpine is absent in the gut of sugar-fed mosquitoes and present in blood-fed ones. This can be normalized to the constitutive expressing lines for instance.

Overall, I don't think that the authors took the opportunity of this revision to improve their manuscript as much as I expected.

Reviewer #3:

Remarks to the Author:

The authors have done a good job at responding to the reviewer's comments. I have no further comments or suggestions and recommend the paper be accepted.

Reviewer #4:

Remarks to the Author:

This is a much improved revision of the original manuscript. This paper will be of wide interest because (to my knowledge) it is the first report of an inducible promoter (blood meal induced) to drive the expression of a antiplasmodium effector protein in a naturally occurring symbiont within mosquitoes. This is an important first step for the use of naturally occurring symbiotic bacteria in the field to counter Plasmodium transmission. I think that it is worthy of publication in Nature Comm.

With that said, there is a responsibility to note to the readers that field applications of this technology would require moving away from a plasmid based approach (necessitating antibiotics for maintenance) towards chromosomal based constructs. Further, the effects towards fitness (both on the host and the homeostasis of the microbiota) are also a logical next step. This was brought up by a couple of reviewers, and I think that it worth a mention in the manuscript, I recommend the discussion.

Response to Reviewer's Comments

REVIEWERS' COMMENTS:

Reviewer #1 (Remarks to the Author):

I am surprised about the patronizing tone used in the point-by-point response. My view of the review process is that reviewers are working for free to help authors improve a manuscript. Some phrasing may have been avoided in this context. However, this is just a comment and this review deals solely with the quality of the revision.

The research described in the previous version of the manuscript was good enough to be considered for revision at Nature Communications. After revision, there has been some improvement in the manuscript itself, even though there are still some small errors (e.g. line 137: Figure 4; Line 205: E. Coil; Line 300: BLAST is not a database; Line 395: AVOVA).

Response: We thank you for catching these. They are all corrected.

In terms of experiments, except from a western blot on bacterial cultures, the authors did not follow any advice to improve their experimental design. For instance, although the vertical transmission issue has been raised by 2 reviewers, the authors decided not to perform them. Apparently, this point put light on an important caveat of this study, that bacteria are quickly lost by mosquitoes. Would it have been that much work to create a stable line expressing the transgene? At least, the issue could have been discussed in the revised manuscript.

Response: The original request was to examine the BMI strains vs the constitutive strains over several generations to see if there was improved performance of the BMI strains over that time span. As this reviewer (and another) pointed out, a prediction of increased fitness of the BMI strains is that they would perform better in the long term. The key idea here is that placing scorpine under the control of BMI promoters increases fitness (survival and reproduction) when compared to strains that are always making scorpine. To test this idea properly, one needs to be able to isolate the effects of the promoters on fitness vs other effects on fitness. One way to make transgenic bacteria is to use transposons. In fact, my laboratory developed the *Himar1 mariner* transposon system that is widely used in many non-model system bacteria and we could have used that in *Asaia*. However, *Himar1* (and Tn5, which was used to make the DsRed strains of *Serratia* that were followed in a similar experiment, see Wang et al. Science. 2017 Sep 29;357(6358):1399-1402) creates random insertions (actually, insertions at "TA", which might as well be random). In a gene-dense bacterial genome, that guarantees that some gene will be disrupted, most likely a different one with each insertion (*Himar1* is widely used for transposon-tagging and mutant screening in bacteria for just this reason). We would have had to make 5 different transgenic lines, none of which would have been inserted into the same site in the chromosome. This introduces a second source of variation in fitness, impossible to disentangle from the variation in fitness between BMI and constitutive promoters which is what we would have liked to measure. The proper way to do this experiment would be to create chromosomal

insertions at the same site in the bacterial chromosome, something that we CAN'T currently do in *Asaia*. We are working hard to solve that problem.

We added a section in the discussion that addresses the shortcomings of plasmid-based *Asaia* paratransgenic strains, noting that they are not suitable for field release.

Considering scorpine expression in the blood-fed mosquito, normalizing to the number of bacteria (1) would not be difficult and (2) would not even be required as we just need to see whether scorpine is absent in the gut of sugar-fed mosquitoes and present in blood-fed ones. This can be normalized to the constitutive expressing lines for instance.

Overall, I don't think that the authors took the opportunity of this revision to improve their manuscript as much as I expected.

Response: The current configuration of the transgene constructs expressed in our strains uses a secretion signal-scorpine-alkaline phosphatase fusion. We detect this in ELISA and on westerns using an anti-*E.coli* alkaline phosphatase antibody. When we use this antibody in westerns where we run blood from the blood bolus, there is always a very high background. It is actually impossible to load enough material on the gels in order to see the scorpine-AP fusion separated from the background binding of this antibody. The next step we would need to take would be to make clones that contained some sort of epitope tag so we could try a different antibody. We chose not to do this since we feel that we have already demonstrated adequately that our constructs are responsive to blood given that 1) the BMI promoters drive expression of GFP in blood fed midguts and not sugar fed ones; 2) the BMI promoters drive expression of the scorpine-AP fusions in the presence of blood on chocolate agar plates but not minimal media; and 3) the BMI strains are strongly antiplasmodial confirming expression of scorpine in blood fed midguts in a biologically-meaningful way.

Reviewer #3 (Remarks to the Author):

The authors have done a good job at responding to the reviewer's comments. I have no further comments or suggestions and recommend the paper be accepted.

Response: Many thanks.

Reviewer #4 (Remarks to the Author):

This is a much improved revision of the original manuscript. This paper will be of wide interest because (to my knowledge) it is the first report of an inducible promoter (blood meal induced) to drive the expression of a antiplasmodium effector protein in a naturally occurring symbiont within mosquitoes. This is an important first step for the use of naturally occurring symbiotic bacteria in the field to counter Plasmodium transmission. I think that it is worthy of publication in Nature Comm.

With that said, there is a responsibility to note to the readers that field applications of this technology would require moving away from a plasmid based approach (necessitating antibiotics for maintenance) towards chromosomal based constructs. Further, the effects towards fitness (both on the host and the homeostasis of the microbiota) are also a logical next step. This was brought up by a couple of reviewers, and I think that it worth a mention in the manuscript, I recommend the discussion.

Response: We added a section on the specific nature of these strains in the discussion that emphasizes why they are not ready for field release. We noted that, in their current form, they are plasmid-based and therefore cannot be maintained stably without selection, that plasmids are prone to horizontal transfer which we don't want to happen, and that we are only expressing a single effector in these experiments, something that is not desirable in field strains since parasites would almost certainly evolve resistance to the single effector very quickly.